Non-destructive detection of highway hidden layer defects using a ground-penetrating radar and adaptive particle swarm support vector machine

Liu Xinyu lxy@ncwu.edu.cn 1 2 3
Hao Peiwen 1
Wang Aihui 4
Zhang Liangqi 3
Gu Bo 2
Lu Xinyan 2
1 CHANG’AN University , Xian , China
2 School of Electric Power, North China University of Water Resource and Electric Power , Zhengzhou , China
3 Henan Wanli Road & Bridge Group Co. Ltd. , Xuchang , China
4 Zhongyuan University of Technology , Zhengzhou , China
Liu Pengcheng
Electronic publication date: 2021 Mar 30
Publication date: 2021
Volume: 7
Electronic Location ID: e417
Received 2020 Dec 8; Accepted 2021 Feb 7
Copyright: ©2021 Liu et al.
Copyright year: 2021
Copyright holder: Liu et al.
License: This is an open access article distributed under the terms of the Creative Commons Attribution License, which permits unrestricted use, distribution, reproduction and adaptation in any medium and for any purpose provided that it is properly attributed. For attribution, the original author(s), title, publication source (PeerJ Computer Science) and either DOI or URL of the article must be cited.
License URL: https://creativecommons.org/licenses/by/4.0/

Keywords: 5 Ground penetrating radar (GPR), Image segmentation, Feature extraction, Support vector machine (SVM), Grid search method, Particle swarm optimization (PSO)

Funding: Scientific Research Projects of Higher Education Institutions in Henan Province 21A120006 Henan Key Youth Teacher Research Project 2016GGJS-074 This work was supported by the Key Scientific Research Projects of Higher Education Institutions in Henan Province (No.21A120006) and Henan Key Youth Teacher Research Project (2016GGJS-074). The funders had no role in study design, data collection and analysis, decision to publish, or preparation of the manuscript.

==============================
In this paper, a method that uses a ground-penetrating radar (GPR) and the adaptive particle swarm support vector machine (SVM) method is proposed for detecting and recognizing hidden layer defects in highways. Three common road features, namely cracks, voids, and subsidence, were collected using ground-penetrating imaging. Image segmentation was performed on acquired images. Original features were extracted from thresholded binary images and were compressed using the kl algorithm. The SVM classification algorithm was used for condition classification. For parameter optimization of the SVM algorithm, the grid search method and particle swarm optimization algorithm were used. The recognition rate using the grid search method was 88.333%; the PSO approach often yielded local maxima, and the recognition rate was 86.667%; the improved adaptive PSO algorithm avoided local maxima and increased the recognition rate to 91.667%.

Introduction

Many forms of road deterioration can develop after prolonged utilization of expressways; examples include crack formation, development of voids, and subsidence (Marecos et al., 2017). The main cause of these conditions is the appearance of cracks under the roadbed, which gradually affects the road surface and causes surface cracks. Continued use of expressways causes incremental damage and can significantly increase the amount and type of maintenance work. With increasing traffic volumes in China, it is necessary to develop more efficient and automated road condition-detection methods.

Much research has been conducted on using ground-penetrating radars (GPRs) for landform surveys. With the continuous development of the GPR technology and with the improvement of detection accuracy, the use of the GPR technology for non-destructive detection of structural road conditions has been garnering increasing attention. Yuanlei et al. used the GPR technology to detect different physical anomalies, based on physical simulations and field tests, and the response characteristics across the two scenarios were consistent (Si & Wang, 2018). Shili et al. used the GPR technology to detect hidden defects, such as road breaks, voids, and subsidence, and acquired defect images (Guo, Xu & Li, 2018). Literature (Liang & Su, 2001) uses amplitude attenuation method in ultrasonic testing to evaluate the corrosion damage of reinforced structures in concrete roads. Document (Mandal, Tinjum & Edil, 2016) detects the defects in concrete road base by ultrasonic detection technology and establishes the mathematical model of sound wave propagation in different media. Literature (Wen et al., 2011) establishes the mathematical model of ultrasonic wave propagation in asphalt pavement and concrete pavement to identify highway surface defects, analyzes its feasibility and realizes ultrasonic nondestructive testing of road defects based on this.

Using methods from the image recognition field, this study focuses on the design of shallow hidden defect classifiers based on the support vector machine (SVM) algorithm. The SVM method has been widely studied and applied in different contexts. Hou improved the SVM algorithm’s low precision around the hyperplane and reduced its computational complexity for processing large amounts of data; they also improved the algorithm’s training efficiency, and managed to reduce the number of false calls (Hou et al., 2018). El-Saadawi & Hatata (2017) used the SVM algorithm for the stator winding protection of synchronous generators, and achieved good results. The SVM algorithm has been widely used in a variety of context, such as big data, medical, agricultural, and transportation applications (Wang, Du & Wang, 2019; Wang et al., 2019; Zhang, Hu & Mao, 2008). Using the SVM algorithm, this study optimizes and improves its parameter selection process. By comparing the optimization performance of the grid search (GS) (Liu & Zhang, 2019) method and the particle swarm optimization (PSO) (Yang et al., 2018) algorithm, the superiority of the PSO algorithm is demonstrated, and the SVM classification algorithm for PSO-based parameter optimization is studied (Ma et al., 2018). By collecting radar images of three diseases on the 107 national highway section from Zhengzhou to Xinxiang in Henan Province, it is shown that the obtained method performs well in detecting hidden pavement defects.

Image preprocessing and feature extraction

Detection principle of the ground penetrating radar

In the GPR approach, the ground is irradiated by high-frequency electromagnetic waves using a transmitting antenna, while the waves reflected from the ground are detected using a receiving antenna. Electromagnetic waves are reflected differently from different underground media; thus, different waveforms registered by the receiving computer for analysis (Benedetto et al., 2017). Figure 1 shows the GPR detection process.

Figure 1 GPR detection process.

Figure 2 Image segmentation using the Canny operator.

(A) Original image. (B) Gauss filter image. (C) Canny operator processing.

Figure 3 Morphological processing of digital images.

(A) Results obtained using the Canny operator. (B) Results obtained using morphological processing.

Figure 4 Projection segmentation results.

The reflection coefficient of an electromagnetic wave mainly depends on the dielectric constants of the medium in which the wave travels originally and the medium from which the wave is reflected, as given by Eqs. (1) and (2):

(1) v=ε1−ε2ε1+ε2

(2) v=cε1

where, r is the reflection coefficient; ε1 ε2 are the relative dielectric constant of the incident medium and the medium from which the wave is reflected, v is the echo speed, and c is the speed of light in vacuum.

Selection of the ground penetrating radar

The ground penetrating radar used in this study is LTD-2100 ground penetrating radar system developed by China Electronics Technology Group Corporation. The main components of the system are 900 MHz shielded antenna, data acquisition host, computer, data cable, etc.

Image preprocessing

The original images of roads detected by a GPR include the reflection characteristics of road conditions and various types of clutter caused by environmental factors. The clutter distribution is typically non-uniform, affecting the correct recognition and classification of road conditions. To improve the accuracy of the road condition determination, it is necessary to remove the effect of this clutter (Bu, 2017). To extract the features associated with various road conditions, the acquired images should be segmented.

Image filtering.

The objective of image filtering is to minimize the effects of noise and interference in raw images. The noise and interference are mainly attributed to the processes –image acquisition and image transmission. Gaussian filtering is typically used for image denoising, which can remove unnecessary interference and protect the edge of the image.

Image segmentation.

The objectives of image segmentation are to classify foreground and background pixels, and to determine the organization of foreground pixels (i.e., detect foreground objects) (Fengjun et al., 2018).

The Canny operator has been widely used for image segmentation of radar images, and its segmentation performance after Gaussian filtering is demonstrated in Fig. 2. Figure 2A is the original radar acquisition diagram. Figure 2B is the result after Gaussian transformation. It can be seen that the clutter in the original image is removed. Figure 2C is the disease characteristic waveform diagram extracted after Canny operator. To retain only the necessary information, the image was subjected to the morphological operations of expansion and corrosion; the results are shown in Fig. 3. Figure 3A is the image after Canny operator segmented the disease waveform. Figure 3B is an enhanced disease waveform diagram after the segmented waveform is processed by digital morphology, so as to facilitate the extraction of disease features in the following text. The processed image was segmented and objects were extracted using the vertical projection method, as shown in Fig. 4.

Feature extraction.

Feature extraction should satisfy the following requirements: (1) features should have strong anti-interference ability; (2) features should be insensitive to translation, rotation, and scale transformation of images; (3) features should be insensitive to geometric distortions; (4) distance between similar images should be as small as possible; and (5) distance between different images should be as large as possible. The algorithm for extracting feature vectors should be simple, and the dimensionality of the feature space should not be too high for ensuring the classification performance of the system (Xiao & Liu, 2017). To satisfy the above requirements, the following features were used: area, image complexity, image texture, and seven rectangular features.

In this study, three types of road hazards are studied, namely, road cracking, hollowing, and subsidence. Through simulation and actual images, combined with the classification of expert experience, three types of samples are collected to describe these conditions. Twentynine feature vectors were extracted for each sample, and the obtained set of features is shown in Tables 1, 2 and 3.

Table 1 Gray level co-occurrence matrix characteristics of cracks, voids, and subsidence.

Defect type	crack	
GLCM	0°	45°	60°	90°	
contrast	1.333	2.149	1.436	2.194	
correlation	0.895	0.826	0.893	0.826	
energy	0.724	0.708	0.728	0.702	
homogeneity	0.988	0.962	0.972	0.967	
Defect type	void	
GLCM	0°	45°	60°	90°	
contrast	0.998	1.501	1.273	1.806	
correlation	0.473	0.235	0.335	0.075	
energy	0.644	0.594	0.626	0.579	
homogeneity	0.891	0.865	0.886	0.849	
Defect type	subsidence	
GLCM	0°	45°	60°	90°	
contra	3.813	6.486	5.447	5.924	
correlation	0.414	0.019	0.164	0.104	
energy	0.795	0.750	0.768	0.758	
homogeneity	0.931	0.884	0.902	0.838	

Table 2 Differential statistical matrix characteristics of cracks, voids, and subsidence.

GLDS	mean	contrast	asm	ent	
crack	0.048	289.831	0.916	0.261	
void	0.141	660.909	0.589	2.186	
subsidence	0.121	765.796	0.794	0.5202	

Table 3 Seventh order invariant moment feature.

seven-order Hu moment	crack	void	subsidence	
Φ1	5.076	4.059	4.463	
Φ2	10.591	8.547	9.925	
Φ3	22.646	15.286	19.998	
Φ4	22.974	15.286	22.229	
Φ5	45.786	30.563	19.562	
Φ6	28.343	19.562	28.508	
Φ7	43.134	35.275	38.163	

The dimensions of the different features are very different. Direct use of feature data not only reduces the system performance, but also affects the classification accuracy. To avoid this shortcoming, it is necessary to perform data normalization (Chowdhury et al., 2019). Let the eigenvector of a pattern vector be X = (x1, x2, …, xm). Then the normalized eigenvector xi′ is (3) xij′=0.1+0.8xij−xi,minxi,max−xi,mini=1,2,…,m

where, xi,max, xi,min are the maximum and minimum of {xi(k)|k − 1, …, P}, and P is the overall number of samples in the training set.

Feature selection.

In this study, K − L transformation was used as the feature selection algorithm (Jun, 2016). K − L transformation can take into account different classification information and realize supervised feature extraction. Under the criterion of minimum mean square error, it can obtain an orthogonal transformation matrix A that can map the original feature X′ from high-dimensional space D to low-dimensional space vector y′. K − L transformation can retain the data component with the largest variance in the original data and highlight the data difference. Empirical knowledge was used for removing several highly correlated features, and K − L changes were used for reducing the dimensionality of the original feature space. A relatively simple class-center classifier was used for identifying the samples after the K − L dimensionality reduction transformation. The recognition rate is shown in Fig. 5 for the test set.

According to Fig. 5, when the dimensionality of the feature space reaches 12, the recognition rate saturates; thus, the optimal dimensionality of the feature space is 12. K − L transformation can take into account different classification information, realize supervised extraction of features with too high correlation, and then use K − L transformation to extract data information from them, thus achieving the purpose of compressing dimensions and improving recognition rate.

Figure 5 Dependence of the recognition rate on the number of features.

The SVM classifier design

Basic principle of the SVM

SVM is a binary classification algorithm that is trained using the supervised learning paradigm. The algorithm attempts to determine an optimal classification hyperplane, whereby the edge distance between two sample classes and the dividing hyperplane (decision boundary) is maximized. The larger the edge distance is, the more separated the two sample classes are, the stronger the classification robustness, and the better the method’s generalizability (Liu et al., 2018). The hyperplane equation is (4) ωTx+b=0

where ω is the normal to the hyperplane, and x determines the angle with the hyperplane; b is the distance between the hyperplane and the origin. The hyperplane is denoted as (ω, b); then, the distance from a certain point (sample) to the hyperplane (ω, b) is denoted by r: (5) r=ωTx+bω

Let the support vector be 1 and point away from the hyperplane, so that ωTx+b=1, and let the vectors in the two different classes be γ and point away from the hyperplane: (6) γ=2||ω||

For optimal segmentation, we need to find the hyperplane with the largest interval; that is, we need to find the parameters ω and b that maximize γ. According to Eq. (6), only ω−1 should be maximized.

Optimization of the SVM classifier parameters

We used MATLAB (Mathworks, Inc.) to validate the diagnostic accuracy of the SVM-based classifier with respect to the road conditions. We obtained a dataset comprising 100 examples of road cracking, hollowing, and subsidence conditions; 80% of these images were used for training the method, while the remaining 20% were used for testing the classifier.

The performance of any SVM classifier depends on the penalty factor and kernel function parameters. The optimal parameter values are typically determined using the grid search approach, which exhausts the set of possible parameter combinations to determine the optimal combination.

The penalty term c and the kernal function parameters g were considered to grow exponentially in the [2−14, 214] range; at the same time, the step between each cell on the grid was set to 0.5, that is, the growth time-points were 2−14, 2−13.5, …, 214. As shown in Fig. 6, the best classification performance was obtained for log2(c)=7.60, best log2(g) = − 6.60.

Figure 6 Parameter optimization results of the grid search method.

As shown in the retrieval results above, the accuracy of log2(c) between [25, 210] and log2(g) between [2−10, 2−4] was relatively good; thus, the step was set to 0.2 in this range, and the parameters were further optimized. The results are shown in Fig. 7.

Figure 7 Results obtained after refining the step size.

According to Fig. 7, the best classification was obtained for best = 7 and best= −5.2. These optimal values were used as the SVM classifier parameters for validating its classification performance; then, the accuracy of the classifier with these parameters was characterized. The validation results are shown in Fig. 8, and the final validation accuracy rate was 88.333%, the image recognition time is t = 0.630 s.

Figure 8 Test-set classification results after the grid search optimization of the SVM classifier parameters.

Adaptive mutation particle swarm

The results obtained using the grid search approach can meet the desired detection requirements, but because the grid search approach only considers discrete combinations of parameters, the optimal solution can be easily missed. At the same time, the grid search method is exhaustive, and needs to consider all the possible cases; thus, this optimization approach is time-consuming. To improve the accuracy and reduce the computation time, the PSO algorithm is proposed in this section.

The PSO approach.

PSO simulates a flock of birds, which are modeled as massless particles (Hsieh et al., 2018). Each particle has only two attributes: velocity v and position x. Velocity represents the speed of movement, and position represents the direction of movement. Each particle searches for the optimal solution separately in the search space, and records it as the current individual extreme value P; the individual extreme values are shared among all the particles in the entire swarm, and the optimal individual extreme value is determined as the current global optimal solution G of the entire swarm of particles. All of the particles in the particle swarm adjust their speeds and positions according to the current individual extremum P found by themselves and the current global optimal solution G shared by the entire particle swarm. The underlying PSO process is relatively simple, and can be divided into the following steps: (1) initializing the particle swarm; (2) evaluating the particles, that is, calculating the fitness values; (3) searching for individual extrema P; (4) searching for the global optimal solution G; (5) modifying the particles’ speeds and positions (Yang, Wang et al., 2019). The update equations are: (7) vidk+1=ωvidk+c1⋅rand0,1⋅pidk−xidk+c2⋅rand0,1⋅pgdk−xidkxidk+1=xidk+vidk+1

Where ω is the inertia factor, c1 and c2 are the acceleration constants, and rand (0,1) are random numbers in the interval (0,1). pid denotes d the dimension of the individual extreme value of variable i.pgd represents the dimensionality d of the global optimal solution. k represents the current number of iterations.

Figure 9 Fitness curve obtained after the PSO.

The following parameter values were used: c1 = 2, c2 = 2, population size = 20, maximal number of iterations = 200, and cross-validation fold K = 5. The fitness curve for the optimized SVM classifier parameters is shown in Fig. 9.

After iteration 15, the fitness reached its optimal value. At this point, the classifier accuracy was 89.197%, the best c was 0.7596, and the best g was 0.7674. Although the basic PSO algorithm exhibits a good convergence speed and good optimization performance, it can prematurely converge onto a locally optimal solution; as a result, the population will easily stagnate without external pressure.

Variation improvement.

To deal with the premature convergence problem, we utilized the concept of mutations that is often used in genetic algorithms, and incorporated mutations into the PSO framework. For that, we defined a trigger to allow particles to escape locally optimal solutions, thus ensuring a global search (Tong et al., 2018). The population fitness variance σ2 was used for determining whether a local maximum was reached in the iteration process. The population fitness variance σ2 was defined as follows: (8) σ2= ∑i=1Nfi−favgf2

In the above equation, N is the number of particles, f is the normalized calibration factor, fi is the fitness of the first particle, and favg is the average fitness. It can be seen that the larger the value of σ2, the more divergent the particle swarm; conversely, the smaller the population fitness variance, the more convergent the particle swarm is. Values of σ2 close to zero indicate that the particle swarm is approaching the globally optimal solution or is converging onto a locally optimal solution. To avoid premature trapping of the particle swarm in local optima, the swarm is subjected to mutations: (9) xik+1=C⋅rand0,1⋅xik

In the above equation, C is a normally distributed random number in the 0,1 interval and is the variation factor; rand(0, 1) is a random number in the 0,1 range; k is the number of iterations. Mutations alter the particles’ positions and thus allow to escape local optima (Deng et al., 2019).

The fitness curve of the improved adaptive mutation PSO algorithm for SVM-based parameter optimization is shown in Fig. 10. Evidently, the best fitness converges to a local maximum after 17 generations. Mutations allow the particle swarm to escape this local maximum after 97 generations; after that, optimization parameters are determined with higher accuracy. The finally estimated parameter values are: c = 7.751, g = 5.754. The fitness at this global optimum is 91.698%.

Figure 10 Fitness curve obtained after the variant PSO.

Simulation results.

Based on the results in the previous section, the SVM model was constructed using the following parameter values: c = 0.7596 and g = 0.7674, which were determined using the PSO method. The results are shown in Fig. 11.

For these parameter values, the accuracy of the model was 86.667%, the image recognition time is t = 0.590s; however, the recognition accuracy had not improved. The main explanation was the trapping of the particle swarm in a local optimum during the process of parameter optimization. An SVM classifier was then constructed and validated using c = 7.751 and g = 5.754; these parameter values were determined using the adaptive mutation PSO algorithm. The validation results are shown in Fig. 12.

From Fig. 12, the classification accuracy of the SVM classification model with the parameter values determined by the adaptive mutation PSO was 91.667%, the image recognition time is t = 0.615s. This classification accuracy is significantly higher compared with that achieved by the SVM method that uses parameters optimized by the grid search approach.

Figure 11 Validation results of the SVM model, for conventional PSO.

Figure 12 Validation results of the SVM model, for variant PSO.

Conclusions

According to the requirements of automatic recognition of highway hidden layer conditions, this paper proposes an automatic detection and recognition method that uses an SVM with parameters optimized using the adaptive mutation PSO approach. In this method, PSO with mutations is used for parameter optimization. In this study, three different methods were used for parameter optimization: (1) the grid search method, (2) the PSO approach, and (3) the adaptive mutation PSO. MATLAB and Python were used for implementing these optimization methods, and the optimization processes and their results were validated. Compared with the grid search method and the simple PSO approach, the accuracy of the SVM with parameters optimized using mutation PSO was higher, translating into better performance on automatic identification of highway conditions. Our simulation results showed that the classification accuracy of the SVM classifier with the grid search method was 88.333%, the classification accuracy of the SVM classifier with PSO was 86.667%, and the classification accuracy of the SVM classifier with mutation PSO was 91.667%. Thus, the effect of SVM classifier with mutation PSO is obviously better than that of the other two. Compared with the grid search method, the classification accuracy was improved. However, owing to the pre-processing of images and processing of feature-related data, some defect-related information is likely to become distorted, which can affect the accuracy of recognition. If a zero-distortion image processing method can be found, the recognition accuracy will be greatly improved. At present, only cracks, voids and subsidence can be analyzed and studied, while asphalt pavement diseases and defects can be divided into 11 categories and 21 items. In order to better improve the scope of the identification system, further research should be done on other types of defects.

Supplemental Information

Supplemental Information 1 Highway structure inspection

Click here for additional data file.

Supplemental Information 2 West side of motorway

Click here for additional data file.

Supplemental Information 3 East side of auxiliary road on the west side of Hong qi Street

Click here for additional data file.

Supplemental Information 4 Gabor 2

MATLAB is required to open this file.

Click here for additional data file.

Supplemental Information 5 Algorithmslib

MATLAB is required to open this file.

Click here for additional data file.

Supplemental Information 6 Gaborfilter 1

MATLAB is required to open this file.

Click here for additional data file.

Supplemental Information 7 Radar image texture feature extraction

MATLAB is required to open this file.

Click here for additional data file.

Additional Information and Declarations

Competing Interests

Author Contributions

Data Availability

Liangqi Zhang is a postdoctoral supervisor in the postdoctoral workstation of Wanli Rord & Bridge Group Co. Ltd. Xinyu Liu is engaged in related research at this postdoctoral workstation. Liangqi Zhang is Xinyu Liu’s second mentor when he is a postdoctoral fellow.

Xinyu Liu conceived and designed the experiments, authored or reviewed drafts of the paper, and approved the final draft.

Peiwen Hao and Liangqi Zhang performed the experiments, prepared figures and/or tables, and approved the final draft.

Aihui Wang performed the computation work, prepared figures and/or tables, and approved the final draft.

Liangqi Zhang performed the experiments, prepared figures and/or tables, and approved the final draft.

Bo Gu and Xinyan Lu analyzed the data, authored or reviewed drafts of the paper, and approved the final draft.

The following information was supplied regarding data availability:

Raw data and code are available in the Supplementary Files.

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
