# Peer review of "Non-destructive detection of highway hidden layer defects using a ground-penetrating radar and adaptive particle swarm support vector machine"

_PeerJ Computer Science, doi:10.7717/peerj-cs.417_

## Round 0.1 · original submission · Major Revisions

This manuscript studies non-destructive detection of highway hidden layer defects using a ground-penetrating radar and adaptive particle swarm support vector machine. It is a topic of interest in using data-driven approaches in non-destructive detection. The reviewers agree that the manuscript has present some interesting findings and step forward from the previous work. However, there are many comments and suggestions as pointed by the reviewers that need to be carefully considered and addressed to improve the manuscript.

Reviewer 1 ·

Basic reporting

no comment

Experimental design

no comment

Validity of the findings

no comment

Additional comments

In this paper, a method that uses a ground-penetrating radar (GPR) and the adaptive particle swarm support vector machine (SVM) is proposed for detecting and recognizing hidden layer defects in highways. This work is interesting. However, the following comments should be cosidered.

(1) Since the main work is to identify the types of road defects, it is necessary to increase the tracking of the progress of the road defect identification content. Plese describe the challenges faced in solving the problem of road defect identification in more details.
(2) In the parameter optimization processing stage, grid search (GS), particle swarm optimization (PSO), and adaptive mutation PSO are used for parameter optimization. It should provide richer and more accurate data for the three methods. Moreover, the authors should highlight the accuracy and superiority of parameter optimization of adaptive mutation PSO.
(3) The paper focuses on the identification of three types of defects, and the comparison of the three parameter optimization algorithms is given. However, it is better to add discussions on other road defects to discuss the effect. Further, the comparision the proposed methods and others methods is suggested.
(4) The literature review and related work are not adequately done.
(5) The K-L transformation is selected when the article selects features. The advange of this method should be added.
(6) When selecting features, which principles are used to select these features, and how do these features help the next step of the recognition work?
(7) More information about the shortcomings of the method and the problems faced in engineering application should be discussed.
(8) Other factors that affect the detection results during the detection should be explained, such as the detection of underground steel bars, which can also reflect echoes, and weather which can afeect the image detection results, and disturbance actors etc.
(9) There are some typos, please check the paper thoroughly. Moreover, it is noted that your manuscript needs carefully editing by someone with expertise in technical English editing paying particular attention to English grammar, spelling and sentence structure so that the goals and results of the study are clear to reader.

Reviewer 2 ·

Basic reporting

1.In the introduction part of the article, you should mention the current application examples of some enterprises or academic institutions in road defect identification, reflecting the current research results.
2.The format of some symbols in the text is wrong and needs improvement.
3.There are grammatical and editorial errors in part of the article. English grammar needs to be improved. You can find professionals whose native language is English to improve.

Experimental design

1.The article provides a comparison of three parameter optimizations, but the processing time and processing results of the three methods are not given in detail. Please add relevant content.
2.When using this method for data processing, which requires higher graphics materials, can the improved method reduce this requirement?
3.There are 100 sets of data samples for training and testing in the article. Are they representative? Will the classification effect be better if the amount of data is increased? I hope the author can try it and display the results obtained in the article.
4.The article does not explain the use of image processing methods, why use Gaussian filtering and K-L transformation, the author can summarize.
5.Mark the type of ground penetrating radar you are using. The data detected by different types of ground penetrating radar are different, which can better show the clarity of the article.

Validity of the findings

1.Why the main discussion direction of this article is the problem of parameter optimization? Explain the author's work in this respect in detail.
2.More discussion and analysis of existing research results can be added to highlight the contribution of the research.

Reviewer 3 ·

Basic reporting

1. In the Abstract, at line 28, the authors use the abbreviation PSO without defining it in advance. It is recommended that the abbreviation should be defined when it is firstly used in the context, as what have been done for GPR and SVM at line 20 and 21.

2. In the Introduction, at line 44, the authors use the term “nondestructive”. However, in the title of this manuscript, it is “non-destructive”. Please format the term.

3. The resolution of the figures in this manuscript looks a little bit low. Could the authors replace them with clearer graphs? For those which are generated from MATLAB, it is recommended that the user use “Export” option in MATLAB to obtain the graphs, which would lead to clearer results, instead of using other snipping tools.

4. In the Image segmentation section, at line 93, the authors state that the image segmentation by Canny operator is demonstrated in Figure 2. While in Figure 2, both the result of image filtering in previous section and the result of Canny operator are shown. Please clarify this in the context to give an exact and complete description of Figure 2. Moreover, could the authors explain the difference between the two results (or among the three images) very briefly?

5. At line 93, the authors use the words “To retain only the necessary information”. Could the authors clarify this in a bit more detail? For example, what is the necessary or unnecessary information, or what is the difference between the two results in Figure 3?

6. Regarding the Figure 4 (C) in the manuscript, please clarify the units of the axes on the plots. Please briefly mention and explain the plots in the context. Please also clarify the units on other plots if applicable.

7. At line 107, it seems that the word “both” should be “all” here, or “both using” should be “using both”? (Grammar)

8. At line 174, regarding the sentence “Speed represents the speed of movement”, it is recommended that the first “Speed” could be replaced by “Velocity” in accordance with the word usage in the previous sentence so that the context would be clearer. (Or vice versa, replace the word Velocity with the word Speed in the previous sentence.)

Experimental design

1. Equation (4) is the same as equation (3). It seems that equation (4) is given incorrectly and should be replaced by another one. Please check the equation.

2. In the Optimization of the SVM classifier parameters section, at line 154, the authors introduce two parameters c and g in the context. Please define or explain the two parameters here.

Validity of the findings

no comment

---

## Round 0.2 · accepted · Accept

The manuscript has been well-revised to meet the satisfaction of the reviewers.

Reviewer 1 ·

Basic reporting

No comment.

Experimental design

No comment.

Validity of the findings

No comment.

Additional comments

The revision is satisfied and the paper can be accepted.

Reviewer 2 ·

Basic reporting

All of the questions were answered.

Experimental design

Every question is responded in this part.

Validity of the findings

The simulation results are given to support validity of the findings.

Reviewer 3 ·

Basic reporting

no comment

Experimental design

no comment

Validity of the findings

no comment

Additional comments

The authors had responded to my previous comments carefully. The revised manuscript is organised very well which provides significant results in defects detection of highway hidden layers.

Only some small issues to add:

(1) In the Introduction section, from line 45 to 55 in the revised manuscript, the authors provide a literature review in this part. However, when reviewing references [2] and [3], the authors use the simple past tense. While in the part of [4]-[6], the simple present tense is used. Similar problem exists in the next paragraph. Please check the literature reviews and use the same form of the tenses.

(2) Similar to above, when describing the authors' designs and simulations, sometimes the authors use the present tense, such as at line 124, 189, 251, 265, etc. While in many other descriptions, the authors use the past tense. Please check the whole manuscript again to correct the forms of the tenses.